# Power fingerprint identification based on the improved V-I trajectory with color encoding and transferred CBAM-ResNet

Lin Lin[1]*, Jie Zhang[1], Xu Gao[2], Jiancheng Shi[1], Cheng Chen[1], Nantian Huang[3]

**1** College of Information and Control Engineering, Jilin Institute of Chemical Technology, Jilin, China, **2** State Grid Hulunbuir Power Supply Company, Hulunbuir City, Inner Mongolia, China, **3** School of Electrical Engineering, Northeast Dianli University, Jilin, China

* jllinlin@126.com

**Data Availability Statement:** All relevant data are within the paper and its Supporting information files.

## Abstract

In power fingerprint identification, feature information is insufficient when using a single feature to identify equipment, and small load data of specific customers, difficult to meet the refined equipment classification needs. A power fingerprint identification based on the improved voltage-current(V-I) trajectory with color encoding and transferred CBAM-ResNet34 is proposed. First, the current, instantaneous power, and trajectory momentum information are added to the original V-I trajectory image using color coding to obtain a color V-I trajectory image. Then, the ResNet34 model was pre-trained using the ImageNet dataset and a new fully-connected layer meeting the device classification goal was used to replace the fully-connected layer of ResNet34. The Convolutional Block Attention Module (CBAM) was added to each residual structure module of ResNet34. Finally, Class-Balanced (CB) loss is introduced to reweight the Softmax cross-entropy (SM-CE) loss function to solve the problem of data imbalance in V-I trajectory identification. All parameters are retrained to extract features from the color V-I trajectory images for device classification. The experimental results on the imbalanced PLAID dataset verify that the method in this paper has better classification capability in small sample imbalanced datasets. The experimental results show that the method effectively improves the identification accuracy by 4.4% and reduces the training time of the model by 14 minutes compared with the existing methods, which meets the accuracy requirements of fine-grained power fingerprint identification.

## 1 Introduction

Residential household customers' electricity consumption continues to rise, and improving residential electricity consumption is of great significance to improving the efficiency of electrical energy utilization [1]. Providing residents with fine-grained energy usage information through Non-Intrusive load decomposition (NILD) has become an important way to effectively reduce energy waste [2]. NILD analyzes the operating status of each appliance from the

**Funding:** This work was supported by Natural Science Foundation of Jilin Province (YDZJ202101ZYTS189). The funder had no role in study design, data collection and analysis, decision to publish, or preparation of the manuscript.

**Competing interests:** The authors have declared that no competing interests exist.

total load data of residential customers which can guide customers to improve their appliance usage habits and reduce household electricity consumption [3–5]. At the same time, it helps the grid to build a more accurate model of household electricity consumption, enabling monitoring, control, management, and friendly interaction with electricity-using devices [6].

Power fingerprint identification is a hot issue in the field of Non-intrusive load monitoring (NILM), which relies on power fingerprint features and classifiers to identify different types of devices. Common power fingerprint characteristics typically include voltage, current, harmonics, power, V-I trajectory, etc [7–9]. The V-I trajectory, as the most common power fingerprint feature, represents the voltage and current waveforms of appliances in the image and has been successfully applied to load identification with good results. In [10], the voltage and current sampling data were plotted as two-dimensional V-I trajectories and then mapped to binary gray images using a normalization method. Reference [11] used V-I trajectory features simplified by elliptic Fourier descriptors, and the classification algorithm was a random forest. Reference [12] used the gramian matrix(GM) color encoding method to construct load markers with color-differentiated load signatures. In [13], a backpropagation (BP) neural network and a convolutional neural network (CNN) are used to extract and fuse the features of the load power and V-I trajectories, respectively. Fused composite features were introduced into the classifier to complete load identification. Reference [14] color-coded the V-I traces to synthesize other load characteristics, but the transient power characteristics are missing and the traces do not reflect the transient characteristics of the device. Research on classification models based on V-I trajectory features is becoming increasingly mature, but the following problems still exist: The original V-I trajectory can only convey trajectory shape information, but cannot reflect other information such as the power of the appliance. Moreover, due to the similarity between different kinds of appliances and working principles, the existing models cannot fully extract the V-I trajectory feature information. Therefore, there is still room for further improvement in the accuracy of its load identification.

Nowadays, scientists and researchers used machine learning (ML) and deep learning (DL) models in several applications including agriculture [15, 16], environment [17–20], and power fingerprint identification. Machine learning is often applied to feature extraction and classification of power fingerprints, such as k-nearest neighbors, support vector machines, decision trees, and random forests. These methods are less computationally intensive, but the identification correct rate is lower. Recently, deep learning has achieved good results in the field of power fingerprint identification, such as CNN, RNN, etc. Meanwhile, researchers propose to construct V-I trajectory images with the help of color coding methods to convert power fingerprint identification into an image classification task in which deep learning excels. However, compared to machine learning, deep learning-based classifiers rely on large-scale training data and longer training time, which limits the application of deep learning.

For V-I trajectory classification, the small amount of load data for specific user results in a poorly trained model with low accuracy or low generalization capability [21]. NILD does not have sufficient computational resources to retrain a complex load appliance identification model, which is costly and time-consuming. Training the model requires a large amount of labeled data, which is difficult to achieve in reality. In recent years, transfer learning has been widely used to overcome the limitations of traditional machine learning. Transfer learning does not require the same distribution assumptions for training and test data as traditional machine learning. This avoids the labor and material costs of relabeling the acquired data in traditional machine learning [22]. The main idea of transfer learning is to use the knowledge from the existing source domain and then transfer it to the target domain to complete its classification. Reference [23] introduced transfer learning to NILD and achieved good results for appliance transfer for the same dataset and cross-domain migration for different datasets.

Reference [24] used the AlexNet network for the load identification of V-I trajectory images and improved the accuracy of load identification using transfer learning. Reference [25] follow the pretraining to reduce the computation in modeling training, and enhance the transferability of the model. V-I trajectories as a load feature that can be converted to image representation are promising for use in load identification in NILD domains.

In addition, there is a class imbalance in the NILM dataset, which can have an impact on the classification performance of the classifier. When the classifier is more biased toward majority class samples, then the identification accuracy of minority samples decreases and the model lacks generalization [26]. Usually, simple methods such as oversampling and undersampling are often used to solve the problem of training unbalanced datasets [27]. However, the oversampling methods for minority classes change the distribution characteristics of the minority class samples [28]. The undersampling method for the majority class loses valuable information about the minority class samples [29]. The above-mentioned methods are difficult to deal with the data imbalance of the V-I trajectory image samples. Reference [30] used a data balancing method based on PixelCNN++ and sample information entropy and used it for V-I trajectories. However, PixelCNN++ has disadvantages such as random generation of wrong images and long training time. Re-weighting [31] is a method that assigns different weights to different classes during the training process. It not only improves the learning ability of the model for a few classes but also reduces the learning ability of the model for most classes.

To solve these problems, this paper proposes a power fingerprint identification based on the improved V-I trajectory with color encoding and transferred CBAM-ResNet34. Experimental results on the PLAID [32] dataset show that the method in this paper improves the accuracy of power fingerprint identification to over 97% compared to other methods. Overall, our main contributions are as follows:

1. Use color encoding to add more feature information to the V-I trajectory and improve the identification of similar electrical appliances. It solves the problem of insufficient feature information in single feature recognition devices.

2. The transferred CBAM-ResNet34 model was constructed to fully extract V-I features and was applied to small samples of V-I trajectory samples. The CBAM and model transferred methods effectively improve the extraction capability of V-I trajectories and sufficiently reduce the training time of the model.

3. For the first time, image-level CB loss is introduced to reweight the loss function to obtain better V-I trajectory identification in unbalanced datasets.

The remainder of this study is organized into the following sections: The second part outlines the overall power fingerprint identification process and introduces the method of acquiring color V-I trajectory. Section 3 explains the construction and improvement process of the proposed model. The validity of the method is verified by the PLAID dataset in Sections 4 and 5. The last section summarizes the work of this paper.

## 2 Materials and methods

### 2.1 The power fingerprint identification process

The power fingerprint identification process includes data acquisition, data pre-processing, feature extraction, and load identification. The specific process of power fingerprint identification in this study is shown in Fig 1. First, collect high-frequency voltage and current data of the appliance and extract the original V-I trajectory of the appliance from the voltage and current data. Then, the construction of color V-I trajectory images using color coding. Finally, the

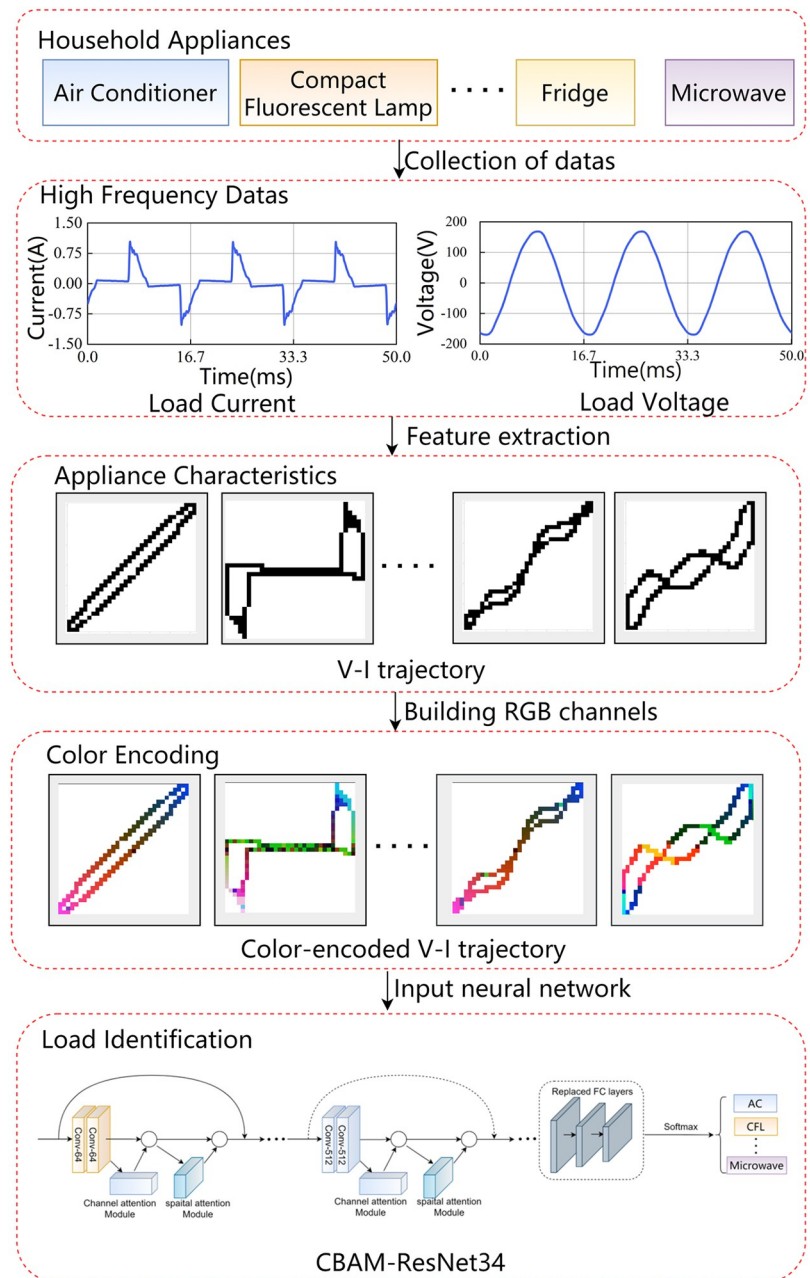

**Fig 1. The process structure of power fingerprint identification.**

color V-I trajectory images are fed into the transferred CBAM-ResNet34 for training. The SoftMax layer classifies the new appliance categories and completes the power fingerprint identification.

## 2.2 Raw data analysis

The primary function of data acquisition is to sample and store the voltage and current waveforms of various household appliances, including (fridges, air conditioners, fans, microwave

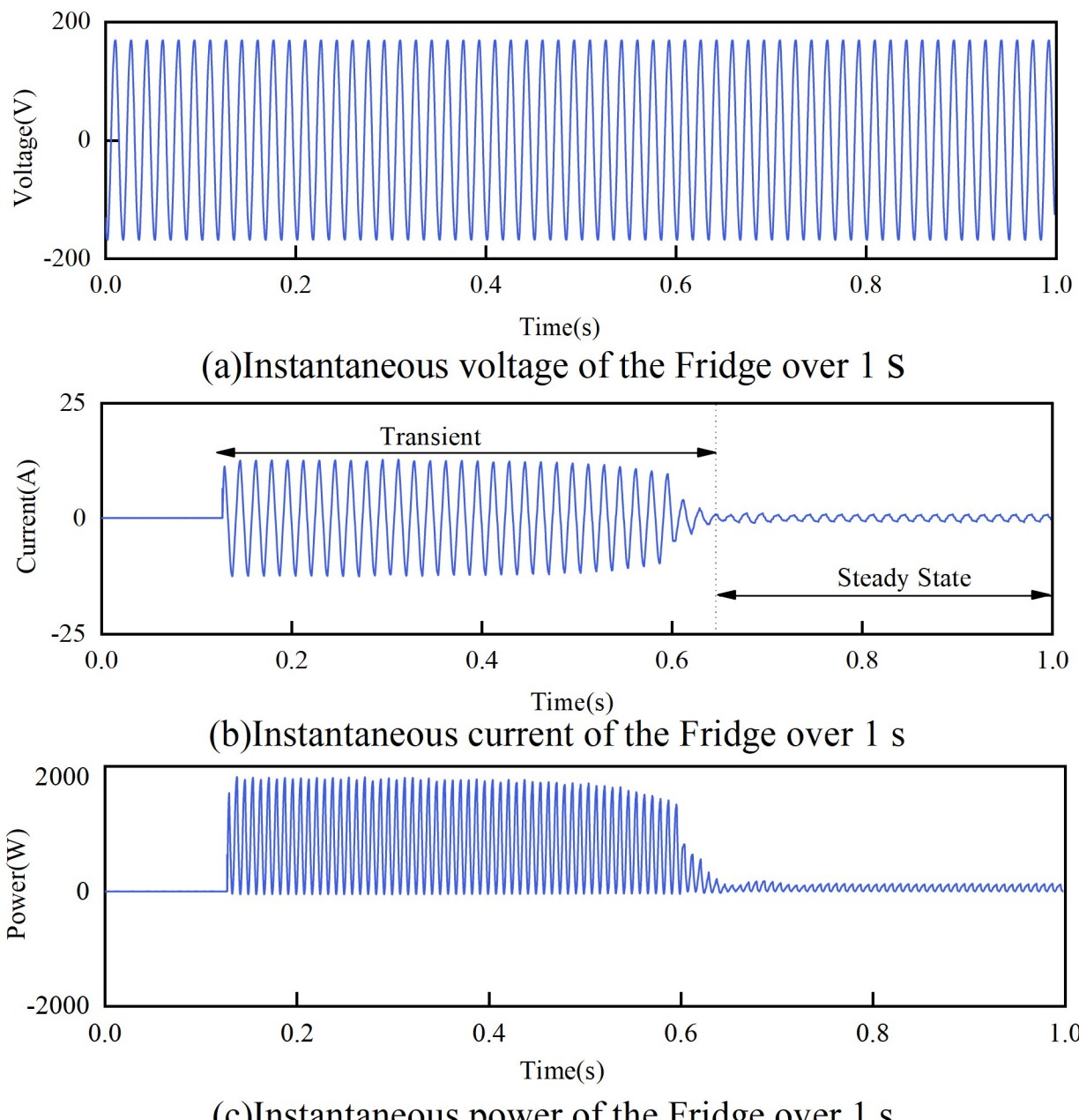

**Fig 2. One instance of a fridge in PLAID.** (a)The instantaneous voltage of the Fridge over 1 s. (b)The instantaneous current of the Fridge over 1 s. (c)The instantaneous power of the Fridge over 1 s.

ovens, etc.) using a collection system [32]. The collection system is usually composed of a current clamp, voltage probe, and high-frequency oscilloscope. The sampling data for the fridge model during the steady-state operation is shown in Fig 2. Therefore, the data are mainly intercepted when the equipment is in steady-state and transient-state operation for feature extraction and energy use analysis.

## 2.3 Construction of color V-I trajectory images based on color coding

The original V-I trajectory image is a single-channel, two-dimensional pixel matrix. Much valuable information is lost in the original V-I trajectory images. In this section, the original V-I trajectory image is combined with the color coding technique to construct a unique power fingerprint label.

First, The V-I traces of different cycles vary slightly due to load fluctuations or noise. When only one waveform cycle is used, this phenomenon inevitably leads to misclassification [24]. Therefore, in this paper, multiple cycles of data are extracted to plot V-I traces to accommodate the dynamic changes in the load.

Then, considering that pixel discontinuity may occur in the mapping process of V-I trajectory images, it is not conducive to subsequent load identification. For this purpose, the traditional mapping method is improved using a bilinear interpolation technique in Fig 3. The specific steps of the bilinear interpolation technique are as follows:

$$D_k = \sqrt{\left(\frac{v_{k+1} - v_k}{\Delta v}\right)^2 + \left(\frac{i_{k+1} - i_k}{\Delta i}\right)^2} > 1 \tag{1}$$

$$v'_{k+t} = v_k + \frac{v_{k+1} - v_k}{T_k + 1}t \tag{2}$$

$$i'_{k+t} = i_k + \frac{i_{k+1} - i_k}{T_k + 1}t \tag{3}$$

where $v_k$ and $i_k$ represent the $k$th data point of $v$ and $i$ respectively. $T_k = \text{floor}(D_k)$ is the number of interpolation points to be added between the $k$th and $(k+1)$th sampling points, floor($\cdot$) represents rounding down. $(v'_{k+t}, i'_{k+t})$ is the $t$th interpolation point of the fill $t = 1,2,\cdots,T_m$.

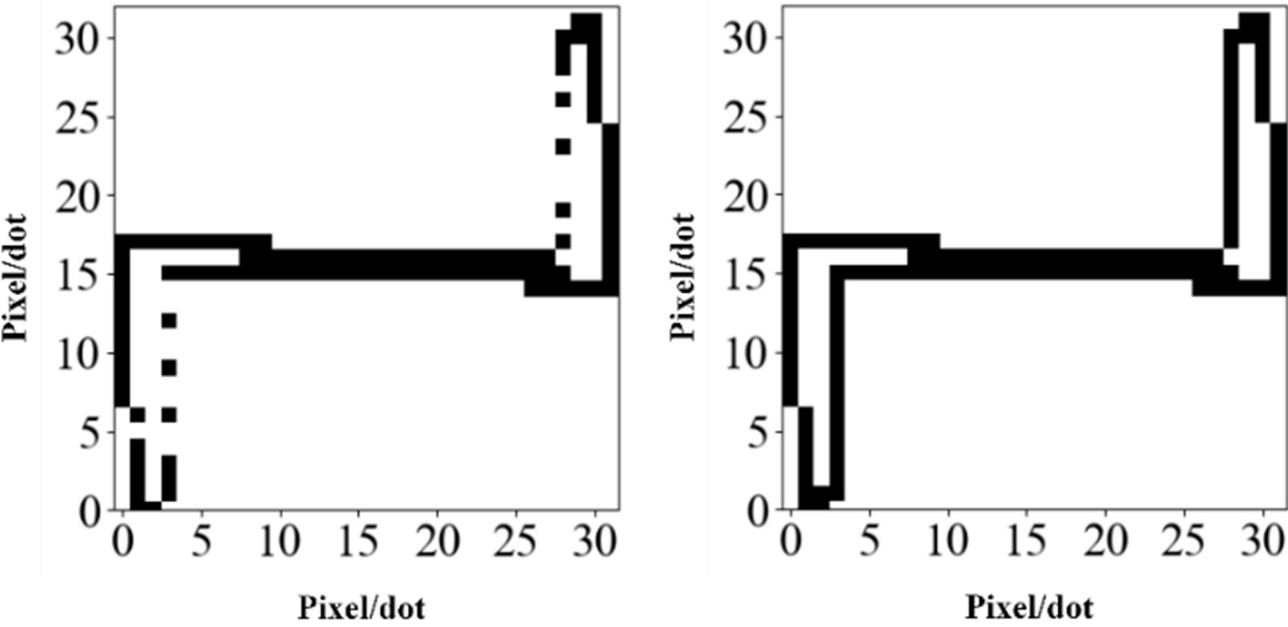

**Fig 3. The binary V- I trajectory mapping for a fluorescent lamp($N = 32$).** (a) Linear interpolation. (b) Bilinear interpolation.

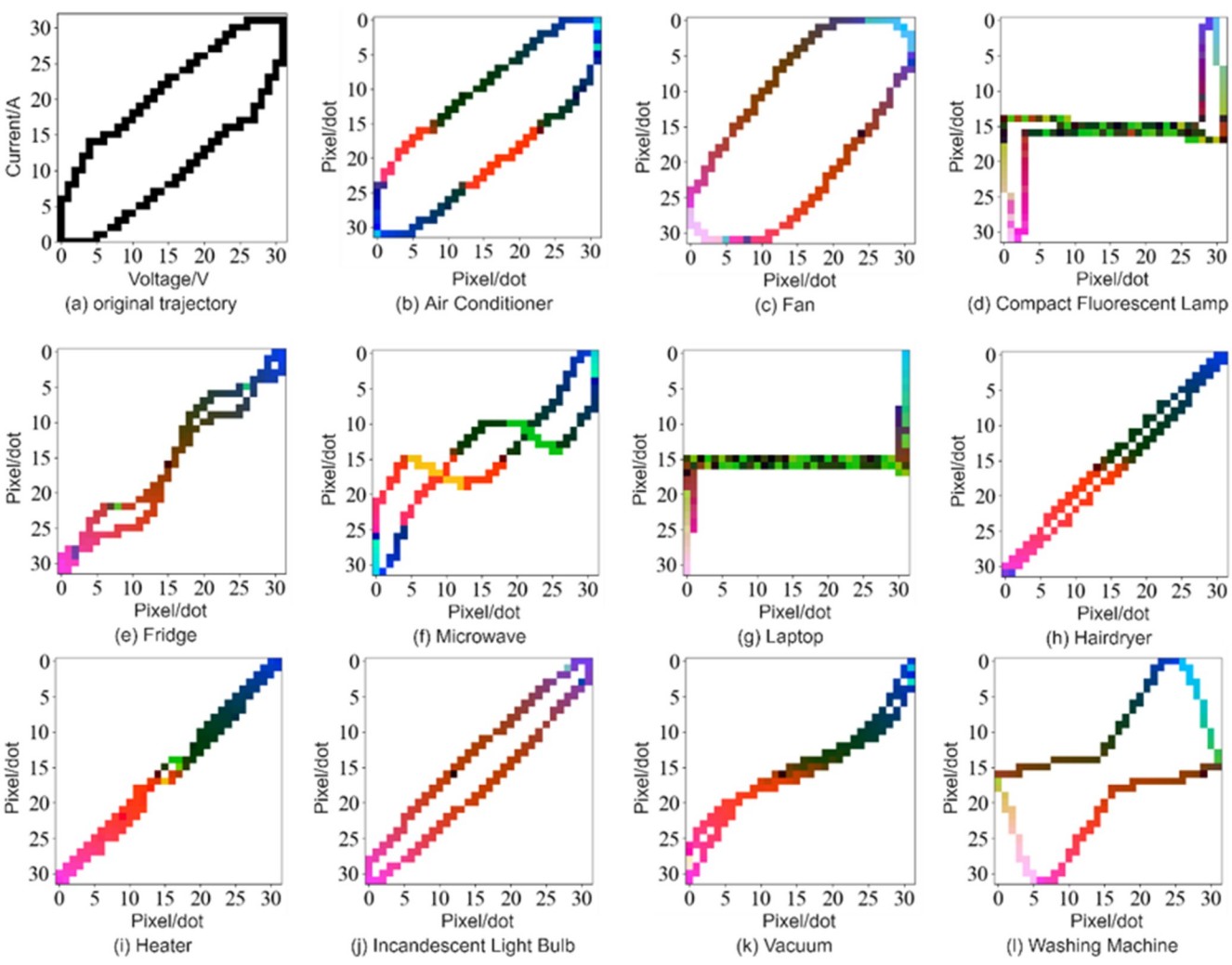

**Fig 4. Color V-I trajectory images of 11 types of appliance loads in the PLAID dataset ($N$ = 32).**

Finally, To cover the load identification information as much as possible with its characteristic parameters, the current, instantaneous power, and trajectory momentum information are added to the original V-I trajectory image using color coding to obtain a color V-I trajectory image in the RGB color space. Each channel corresponds to a two-dimensional matrix, and each matrix element can vary continuously from zero to one. V-I trajectories were plotted for 11 categories of electrical equipment randomly selected from the PLAID dataset as shown in Fig 4. The specific steps of the color encoding method are as follows.

- Red channel: The shape of the V-I trajectory depends heavily on the load current that reflects the physical characteristics of the device. The average current of the device during stable operation is populated in the R channel, aiming to extract a more stable V-I trajectory and ensure the classification effect.

$$R(m) = \left( \left\lfloor \frac{i_m - i_{\min}}{i_{\max} - i_{\min}} \times N \right\rfloor, \left\lfloor \frac{v_m - v_{\min}}{v_{\max} - v_{\min}} \times N \right\rfloor \right) \tag{4}$$

Where $i_{\max}$, $i_{\min}$ are the maximum and minimum values of the current sampling values, respectively; $v_{\max}$, $v_{\min}$ are the maximum and minimum values of the voltage sampling value, respectively; $\lfloor\ \rfloor$ represents rounding down.

- Green channel: The V-I trajectory characteristics in the transient state are somewhat different from those after the stable operation. Therefore, in this paper, the average instantaneous power of the device in the transient state is filled in the G channel. The transient V-I trajectory characteristics are also an important characteristic for different loads.

$$\boldsymbol{G}(m) = \frac{1}{K}\sum_1^K W_k \tag{5}$$

Where $K$ represents the number of cycles of the load in the transient state. $W_k$ represents the corresponding pixel value in the V-I trajectory feature at the $k$th instantaneous state point value.

- Blue channel: the rate of change of voltage and current during the stabilization cycle varies from device to device. V-I trajectory has a loop direction, reflecting the phase relationship between current and voltage. To capture the motion information the of V-I trajectory, the blue channel is plotted by the voltage and current of the appliance for continuous 20 cycles in steady-state operation.

$$\boldsymbol{B}(m) = \frac{\arg\left(\frac{v_{m+1}-v_m}{v_{\max}}, \frac{i_{m+1}-i_m}{i_{\max}}\right)}{2\pi * 20}, m = 1, 2, 3\cdots, M \tag{6}$$

The above color coding steps fused the Red, Green, and Blue channels to obtain the corresponding color V-I trajectory images in RGB spaces. Color encoding increases the uniqueness of the V-I trajectory and provides better identification.

## 3 Power fingerprint identification based on transferred CBAM-ResNet34

### 3.1 CBAM-ResNet34 model

The ResNet model effectively mitigates degradation and gradient disappearance during model training by stacking the residual structures [33]. Networks with fewer layers lack certain feature representation capabilities. On the other hand, the increase in the number of network layers is accompanied by an increase in the number of parameters and computations, which makes the network training slower. However, the residual structure in the ResNet34 neural network has a limited ability to extract V-I trajectory features and cannot ignore irrelevant information in the model training. By applying spatial attention and channel attention, the CBAM module can amplify the weights of effective channels in the feature layer. This enhances the feature representation of V-I trajectories and improves their saliency, laying the foundation for accurate classification of subsequent load recognition.

The CBAM module represents the attention mechanism module of the convolutional module, which is an attention mechanism module that combines spatial and channel [34]. Compared to senet, which only focuses on channels, CBAMs achieve better results. They save parameters and computational power and can be integrated into existing networks as plug-and-play modules.

The details of the channel attention module (CAM) and spatial attention module (SAM) are shown in Fig 5. The CAM focuses more on the more critical parts of the image and ignores

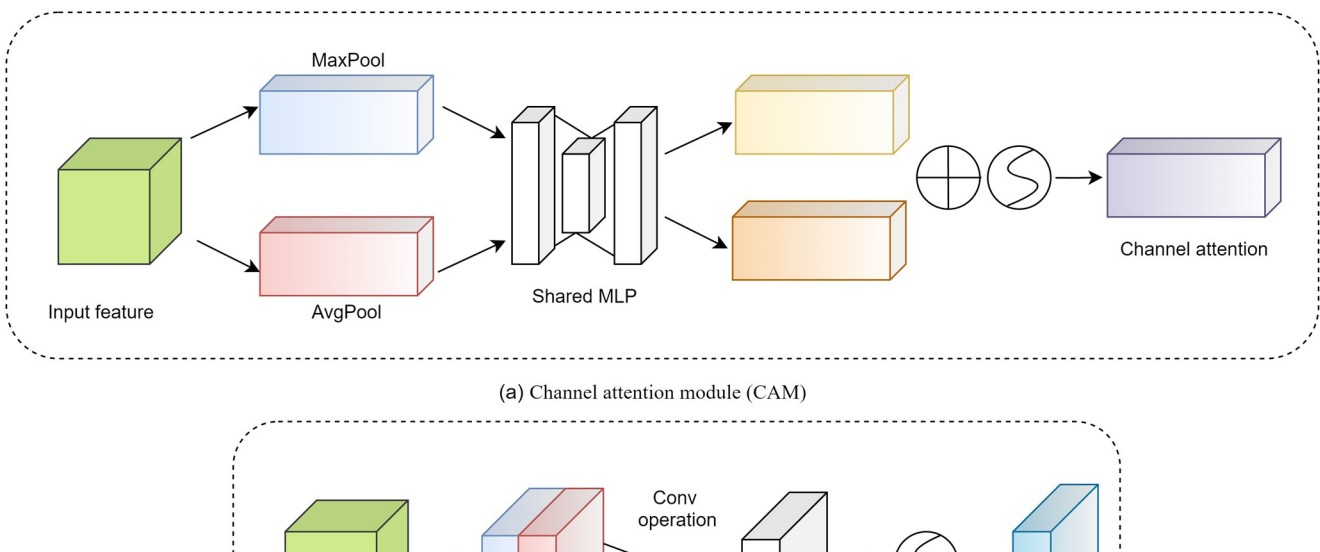

(a) Channel attention module (CAM)

(b) spatial attention module (SAM)

**Fig 5. The structure of the Convolutional Block Attention Module (CBAM) model.**

extraneous information. First, the input features are processed in parallel by averaging pooling and maximum pooling. Subsequently, the multilayer perceptron (MLP) forwards both types of data with a hidden layer. Finally, the output features are merged by element summation.

The CAM can be expressed as follows.

$$\begin{aligned} \mathbf{M}_c(\mathbf{F}) &= \sigma(MLP(AvgPool(\mathbf{F})) + MLP(MaxPool(\mathbf{F}))) \\ &= \sigma\left(\mathbf{W}_1\left(\mathbf{W}_0(\mathbf{F}_{avg}^c)\right) + \mathbf{W}_1\left(\mathbf{W}_0(\mathbf{F}_{max}^c)\right)\right) \end{aligned} \quad (7)$$

where $\mathbf{W_0}$ and $\mathbf{W_1}$ are learnable weights and $\sigma$ is an S-shaped function.

SAM is a complement to CAM and its main purpose is to discover the most meaningful information after CAM processing. First, the input features are processed serially through the mean and maximum pools. This information is then forwarded by the convolutional layer. The final mathematical representation is as follows.

$$\mathbf{M}_s(\mathbf{F}) = \sigma\left(f\left([AvgPool(\mathbf{F}); MaxPool(\mathbf{F})]\right)\right) \quad (8)$$

where $f$ denotes the convolution operation.

In this paper, the position of CBAM is added to each residual block to reduce the influence of redundant features with the help of an attention mechanism and improve the accuracy and timeliness of subsequent recognition. As in Fig 6(a) and 6(b), the activation function of ReLU was used before each weighting layer. After extracting the CNN, the CBAM layer extracts the most critical information through the channel and spatial dimensions. The attention mechanism is more likely to learn effective features in shallow networks, and the improvement effect

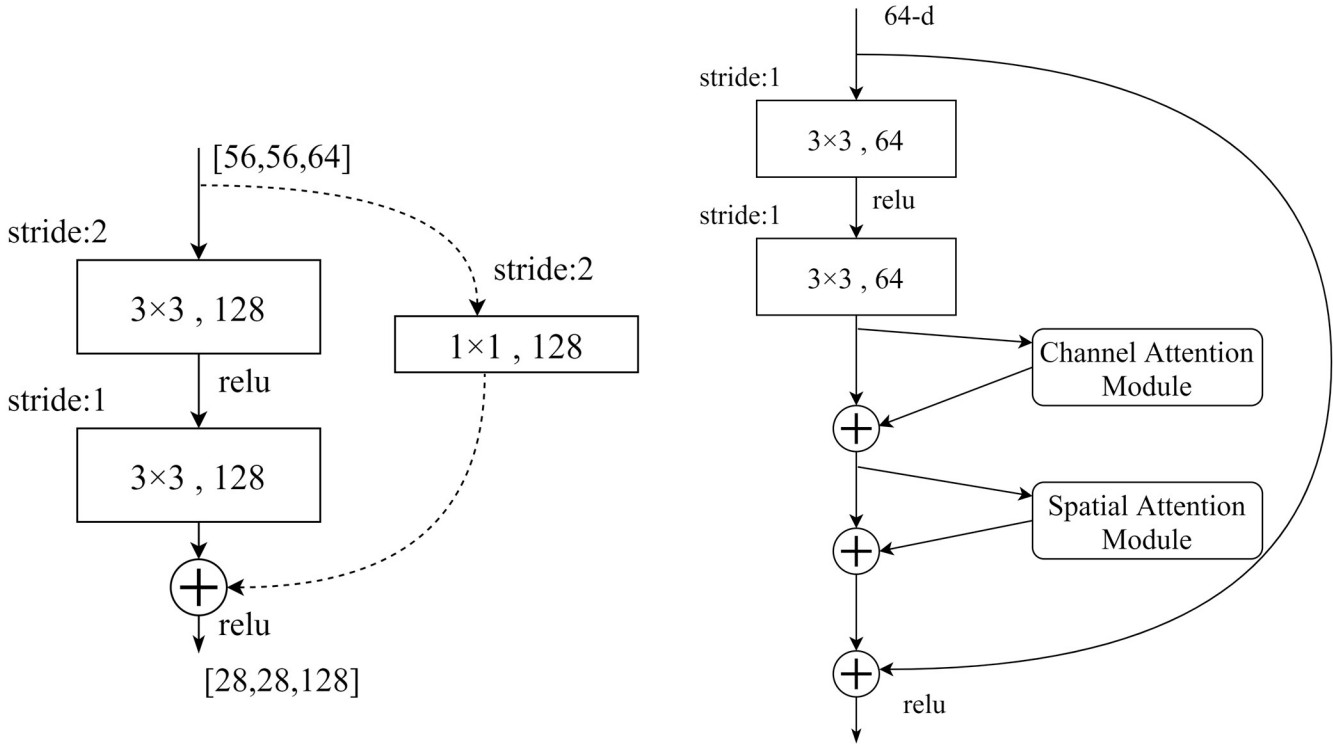

**Fig 6.** (a)The ResNet network residual structure. (b)The structure of the CBAM-ResNet model.

of the attention mechanism module on the performance of deeper networks is less obvious, and it is difficult for the attention mechanism to learn effective features.

## 3.2 Model transfer method

The ResNet model was initially designed to identify the ImageNet datasets. However, the learned high-level abstract features can still be used to assist in identifying the V-I trajectories of different types of appliances. We used a model-transferring method and added the CBAM attention module for the ResNet networks, as shown in Fig 7. The specific training steps were as follows:

- The ResNet34 neural network was pre-trained using the ImageNet dataset, and all layers except the last fully connected layer were extracted from the pre-trained ResNet34.

- The Simple Attention module of CBAM was introduced, which was added to each residual structure module of ResNet34. The last full connection layer was replaced with a new one, and the layers were transferred to the new electrical equipment identification task.

- Input color V-I trajectory image. The color V-I trajectory image size was adjusted to the same size as the input neuron of ResNet34.

- Output the categories of the electric loads. The new full connection layer is adjusted to the same number of electrical equipment identification categories.

- All the parameters are retained until the termination condition is satisfied.

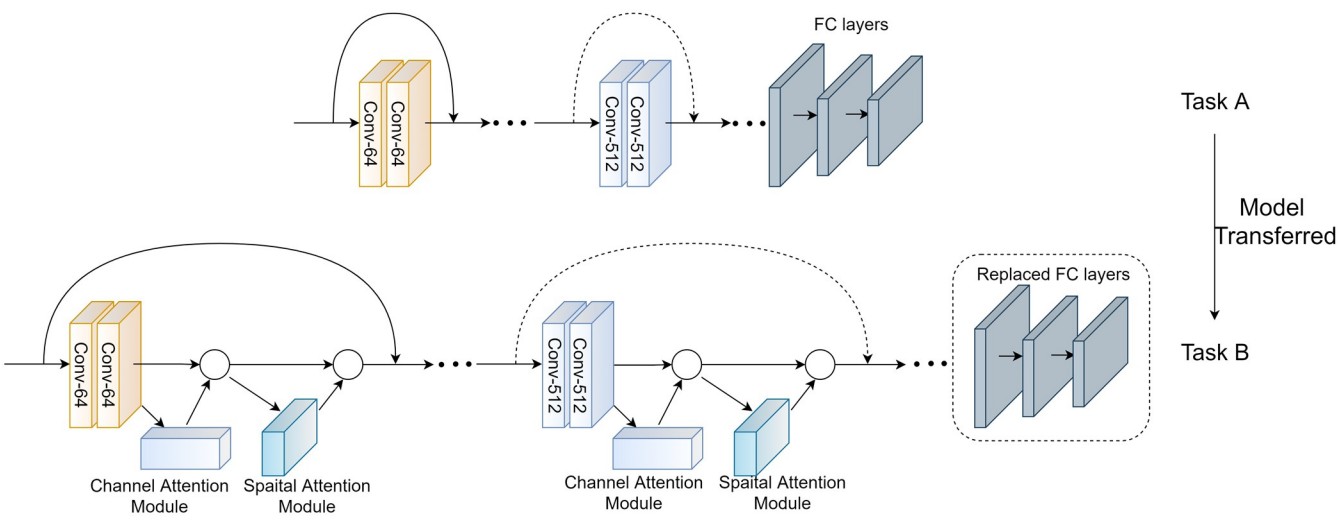

**Fig 7. The experimental scheme of model transferring.**

## 3.3 Class-balanced loss

In power fingerprint identification, data set imbalance is another issue of concern. The usage frequency of household appliances varies greatly, which is reflected in the data set as a significant variation in the sample sizes of different categories. To solve this problem, it is usually necessary to expand some samples so that the number in different categories is the same. The class-balanced loss is an image-level class-balanced loss function that can be implemented by inverse weighting the loss function by the number of effective classes.

Therefore, to improve the identification accuracy of color V-I trajectory images, we rebalance the loss by weighting the loss function using the number of valid samples per class to improve the loss. A weighting factor is introduced in the class-balanced loss [35]. We apply it to softmax cross-entropy loss as follows:

Assume that the predicted output of the model for all classes is $\mathbf{z} = [z_1, z_2, \cdots z_c]^T$, where C is the total number of classes. The softmax function treats each class as mutually exclusive and calculates the probability distribution of all classes as

$$p_i = \frac{\exp(z_i)}{\sum_{j=1}^{C} \exp(z_j)}, \forall i \in \{1, 2, \ldots, C\} \tag{9}$$

Given a sample with the class label $y$, the SM-CE loss for this sample is written as:

$$\mathbf{CE}(\mathbf{z}, y) = -\log\left(\frac{\exp(z_y)}{\sum_{j=1}^{C} \exp(z_j)}\right) \tag{10}$$

Suppose the class $y$ has $n_y$ training samples, the class-balanced SM-CE loss is:

$$\mathbf{CB}(\mathbf{z}, y) = -\frac{1-\beta}{1-\beta^{n_y}}\log\left(\frac{\exp(z_y)}{\sum_{j=1}^{C} \exp(z_j)}\right) \tag{11}$$

The class-balanced cross-entropy loss uses a balancing factor $\omega$ to increase the weight of minority samples in the target loss and decrease the weight of majority samples in the target

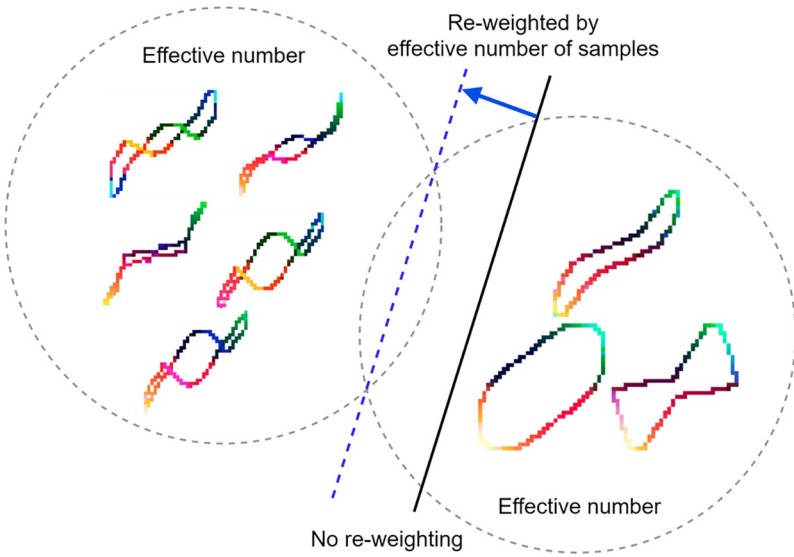

**Fig 8. The schematic diagram of the reweighted optimal classifier model.**

loss, which makes the classifier focus more on the features of minority categories in training, thus improving the category imbalance problem in the dataset and increasing the accuracy of recognition.

The 2 appliances in the PLAID dataset, Microwave (majority class) and Washing (minority class), have different sample sizes as shown in Fig 8. The model trained based on these samples is biased towards the majority sample (black solid line). The class balancing loss (Cbloss) is introduced to reweight the loss by reversing the effective number of samples. The generalization performance of the model can be improved when trained with the proposed class balance loss (blue dashed line).

# 4 Experiments

## 4.1 Dataset selection

In the actual example, Python 3.8 and Pytorch 1.3.0 deep learning frameworks are used in the actual arithmetic example. The hardware platform was GeForceRTX3090, and the software

**Table 1. The PLIAD dataset of appliance types and instance statistics.**

| Appliance Type | Appliances | Instances |
|---|---|---|
| Air Conditioner | 21 | 208 |
| Compact Fluorescent Lamp | 40 | 220 |
| Fan | 30 | 210 |
| Fridge | 30 | 90 |
| Hairdryer | 37 | 248 |
| Heater | 13 | 85 |
| Incandescent Light Bulb | 30 | 148 |
| Laptop | 41 | 207 |
| Microwave | 37 | 229 |
| Vacuum | 15 | 73 |
| Washing Machine | 18 | 75 |
| Total | 312 | 1793 |

platform was Linux OS Ubuntu 18.04. Hardware acceleration using GPUs when training deep learning models. The proposed load identification method was tested using the PLAID dataset. The dataset recorded 1793 sets of voltage and current data of 312 appliance loads in 11 categories from 55 households at a sampling frequency of 30 kHz [32]. The number of class samples and examples of the 11 classes of appliance loads are listed in Table 1. 20% of the data were randomly selected as the test set, and 80% were selected as the training set.

## 4.2 Parameter optimization

The number of iterations and pixels of the color V-I trajectory image was optimized to obtain the best parameters for load identification. When analyzing the influence of one parameter on the identification result, the other parameter was fixed. After determining the input and output of the model transfer, the new network model was trained until the termination condition was reached. The cross-entropy function was chosen as the minimum loss function, and the optimizer was chosen as Adam. The initial learning rate was configured as $1 \times 10^{-4}$ to reduce the learning rate of the transferred layer. Meanwhile, the learning rate of the new fully connected layer was increased to $1 \times 10^{-3}$.

To quantitatively analyze the effect of the number of iterations on the method, the variation curves of the identification accuracy and loss value with the number of iterations during a specific test were plotted, as shown in Fig 9. The number of iterations for the entire test was set as 300. It can be seen that as the number of iterations increases, the identification accuracy increases rapidly, and the loss value decreases. When the number of iterations reaches 200, both the loss value and the identification accuracy stabilize, and the classification performance of the method no longer changes significantly. Based on the above analysis, the number of iterations was set to 200, and a more satisfactory identification result could be obtained in a relatively short period.

Fig 10 shows the training process of the CBAM-ResNet34 model and the transferred CBAM-ResNet34 model that was used to validate the feasibility and effectiveness of the model transferring method. It can be seen that the convergence rate of the transferred CBAM-ResNet34 model is faster than that of the CBAM-ResNet34 model. The pre-trained model maintains a high accuracy at the beginning of training. When the number of iterations reaches around 10, the identification accuracy exceeds 90% and increases slowly in subsequent training. The method of model transferring improves the training speed and accuracy of power fingerprint identification while retaining prior knowledge, as the abstract feature extraction capability of the pre-trained model can be shared across image identification.

A box plot of the load identification accuracy at different resolutions is shown in Fig 11 to select the appropriate pixel size for the color V-I trajectory image. It can be seen that the overall identification accuracy of load recognition is best when the resolution N of the color V-I trajectory image is 128. However, with increasing image resolution, *N*, the identification accuracy gradually decreased. When the image resolution is small, the color V-I trajectory image is not clear enough, and there is insufficient information about the features. When the resolution of the image is increased to a specific range, more feature information is contained, and CBAM-ResNet34 can extract the feature information. In addition, an image resolution that is too high can also increase the effect of random noise and perturbations, resulting in sharp color V-I trajectory images that are not conducive to subsequent feature extraction. This causes the accuracy of the load identification to decrease rather than increase. Given this, the resolution *N* of the color V-I image was set to 128.

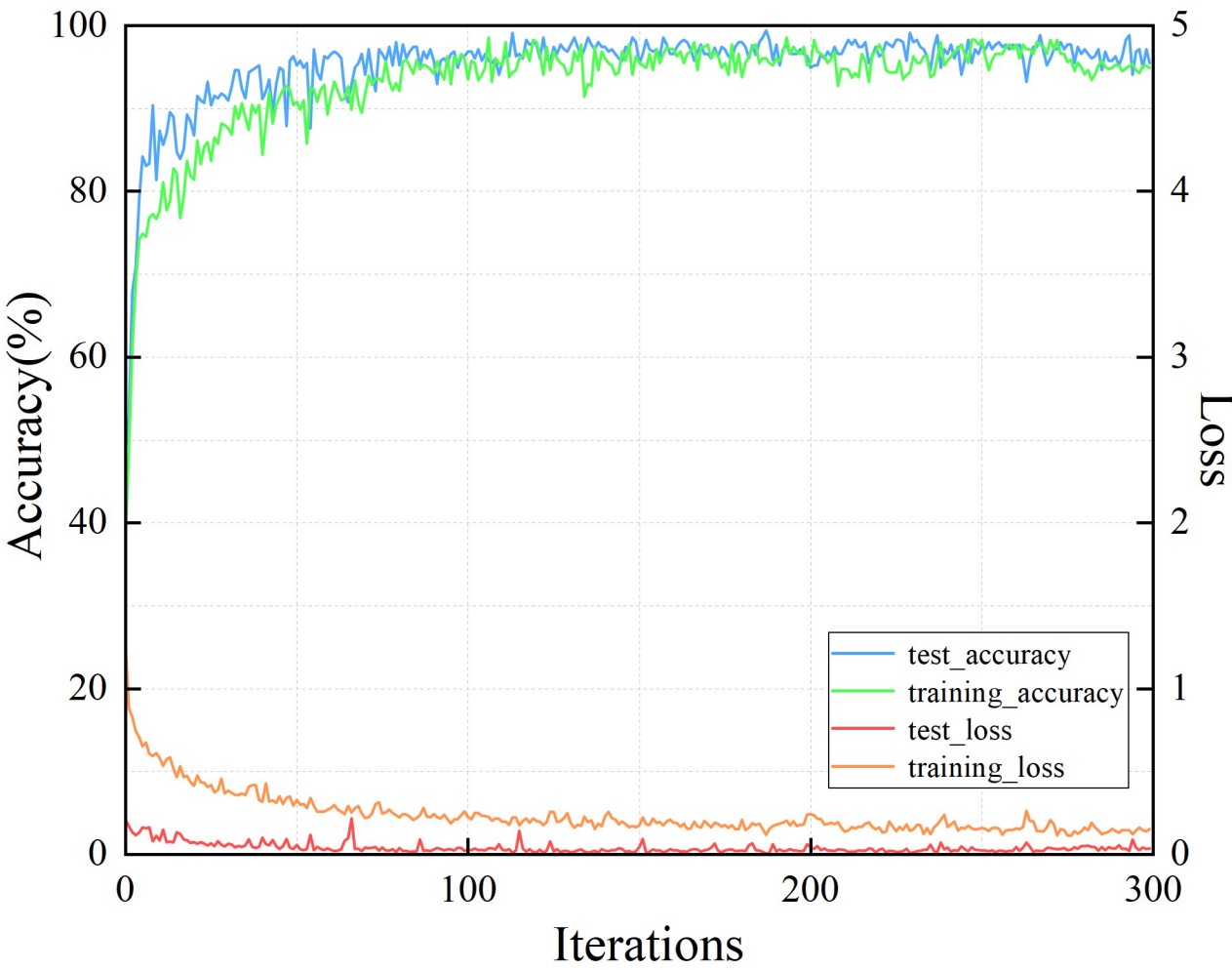

**Fig 9. The comparison of several iterations.**

## 5 Result and discussion

### 5.1 Evaluation metrics

To analyze the classification results of this method more comprehensively rather than simply using classification accuracy for judgment, a confusion matrix [36, 37] is adopted as the evaluation standard for the classification results. The confusion matrix of each comparison classification result through visual analysis of the data is shown in Fig 12. The numbers in the graph represent the current sample size, and the percentages represent the current sample size as a percentage of the total sample size. The evaluation index $P_{re}$ represents *the* precision (shown as a green percentage in the last column in Fig 12). The evaluation index $R_{re}$ represents *the* recall (shown as a green percentage in the last row in Fig 12). $F_{score}$ represents a reconciled average assessment indicator for precision and recall. These are calculated as follows [38, 39]:

$$P_{re} = \frac{T_p}{T_p + F_p} \tag{12}$$

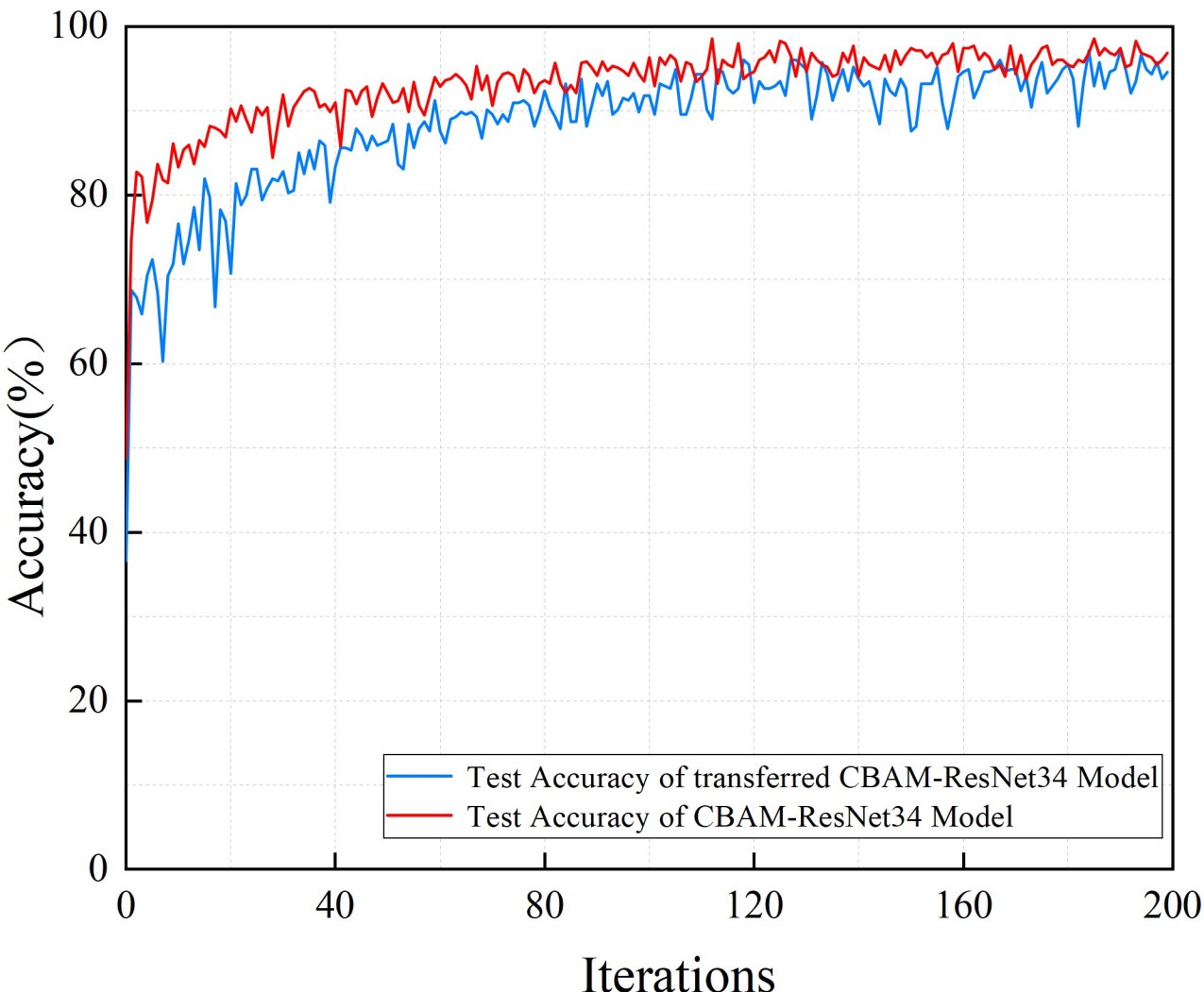

**Fig 10. The training and validation accuracy of CBAM-ResNet34 and transferred CBAM-ResNet34 models are illustrated for comparison.**

$$R_{re} = \frac{T_p}{T_p + F_N} \qquad (13)$$

$$F_{score} = \frac{2P_{re}R_{re}}{P_{re} + R_{re}} \qquad (14)$$

where $T_p$ means that the actual values are valid and classified as positive, $F_p$ means that the real value is false and is classified as positive, and $F_N$ means that the actual values are valid and unfavorable.

## 5.2 The performance of V-I trajectory with color encoding and CBAM

Fig 12(a) and 12(b) show that when color V-I trajectory images were used for load identification simultaneously, the accuracy rate before the model transfer was only 94.4%. In

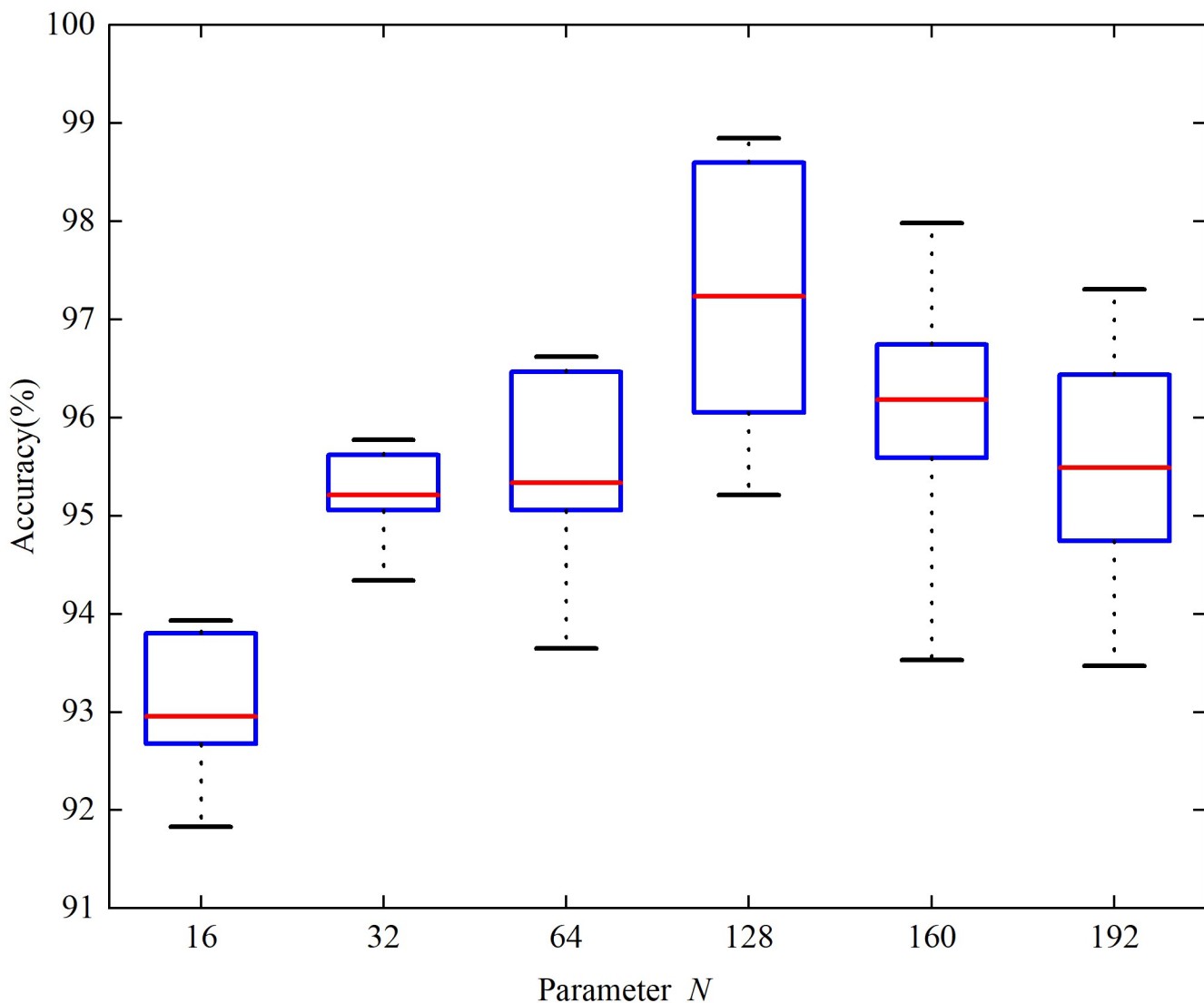

**Fig 11. The selection of parameter N.**

comparison, the accuracy rate after the model transfer was as high as 97.5%. The strategy of model transfer is to realize the transfer from the multi-classification of large datasets in the source domain (such as the ImageNet dataset) to specific learning tasks in the target domain by adjusting and optimizing the structure and parameters of the pre-trained model. Applying the underlying features learned by ResNet34 on large datasets to the V-I trajectory image classification problem reduces the complexity of the model optimization. This enables the network to perform well even in small sample datasets, improving the accuracy of load identification and reducing the training time.

As shown in Fig 13, the F-scores of the air conditioner, washing machine, and heater are lower than $F_{macro}$ (the average of all F-scores of appliance loads). The fridge, air conditioner, and washing machine are multi-state appliances with limited working modes [40]. For example, air conditioners have a variety of working conditions, such as heating and ventilation

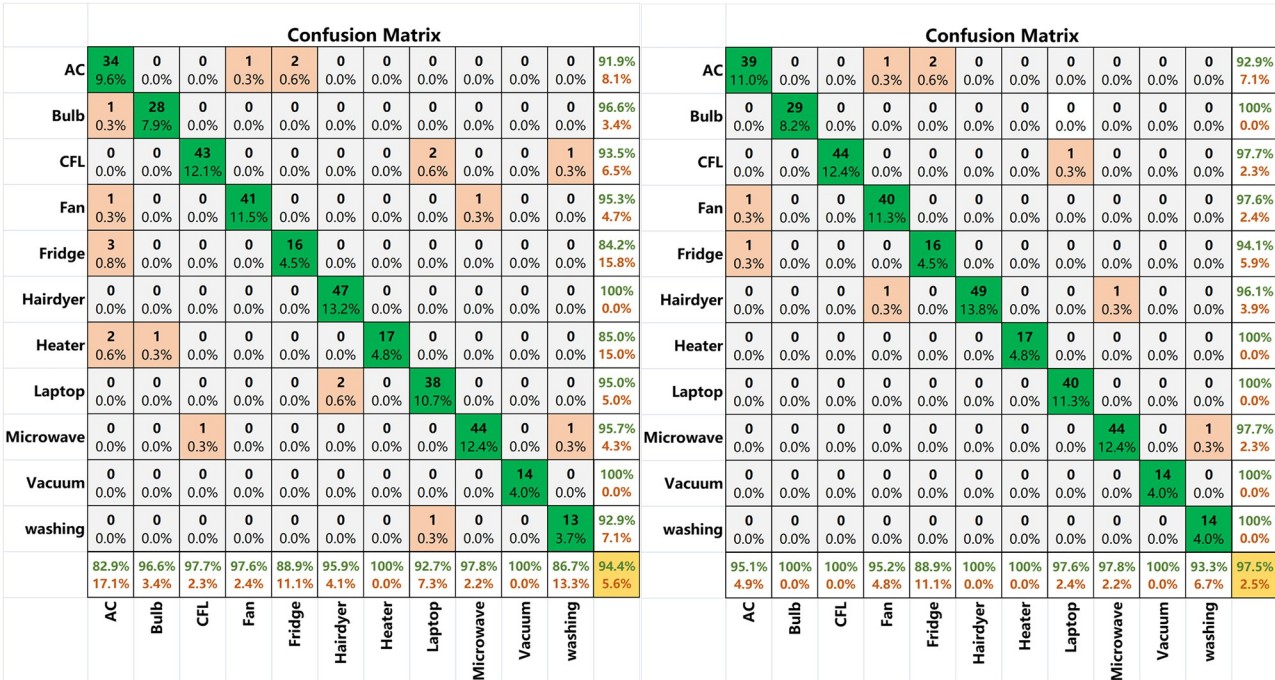

**Fig 12. The confusion matrix for appliance loads of the PLAID dataset.** (a) Confusion matrix based on color V-I trajectory image before model transfer. (b) Confusion matrix based on color V-I trajectory image after model transfer.

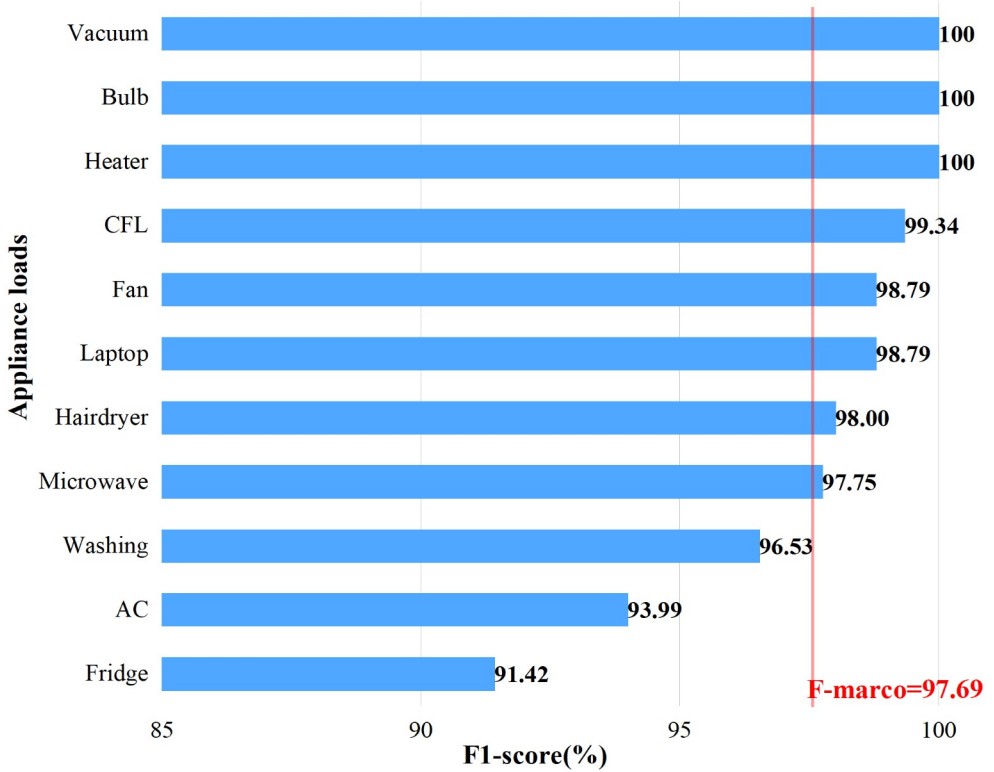

**Fig 13. The F-score (%) for appliance loads of the PLAID dataset.**

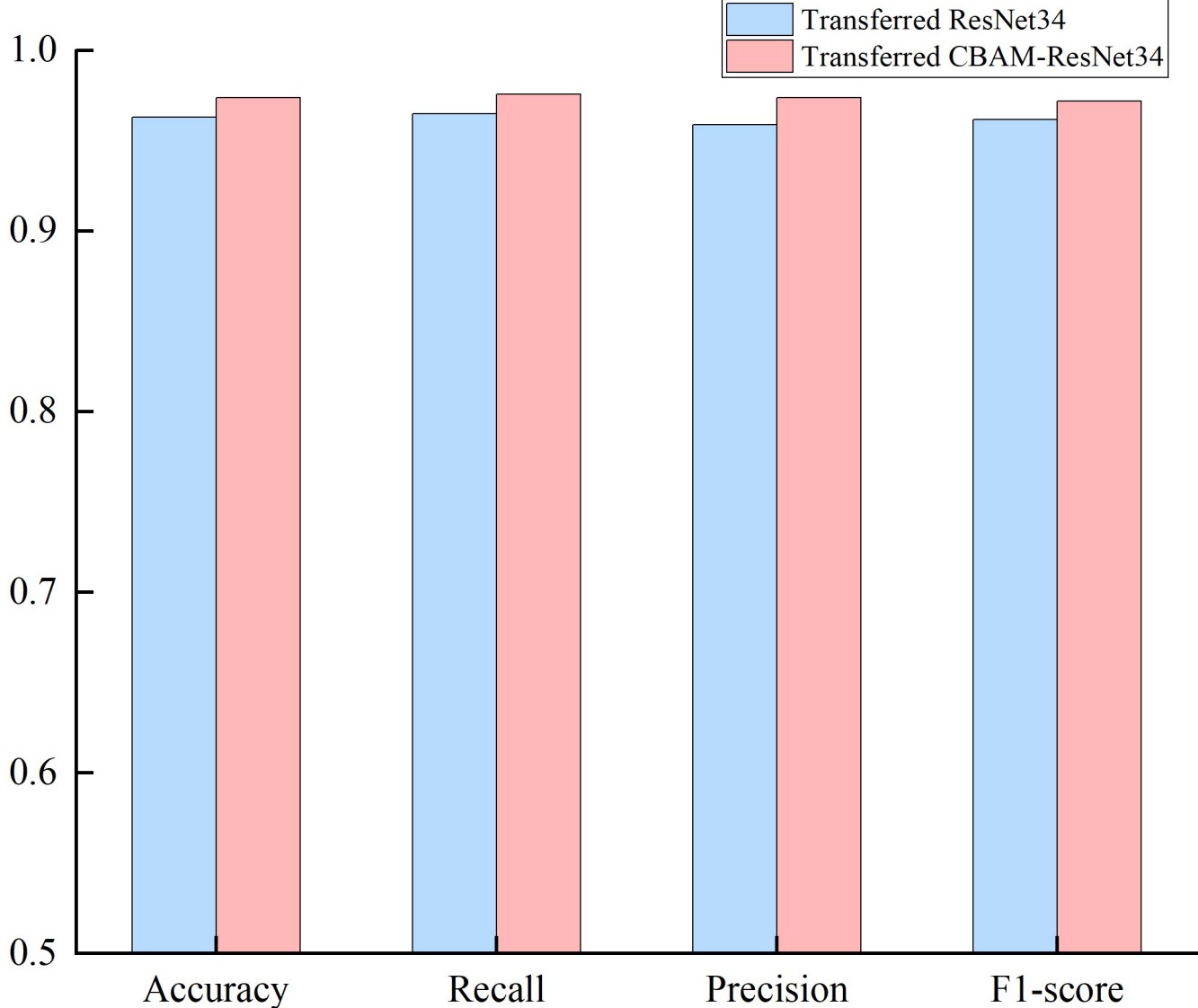

**Fig 14. The evaluated metrics between transferred ResNet34 models with and without CBAM.**

modes, which are easily confused with hairdryers and heating machines. The fridge has a cooling fan and compressor motor, easily be confused with washing machines and fans. Because these appliances switch between multiple operating modes, their electrical characteristics are complex, making load identification more difficult.

To verify the effectiveness of the CBAM module, the classification results of transferred ResNet34 and transferred ResNet34-CBAM are compared on the PLAID dataset. It can be found that the transferred ResNet34 CBAM has better results than the transferred ResNet34 in all five evaluation metrics as shown in Fig 14. The transferred ResNet model with the CBAM module can better classify the images of V-I trajectories formed by 11 different classes of electrical devices compared to the model without CBAM.

**Table 2. Comparison of identification accuracy and training time between different training models on the PLAID dataset.**

| Training Model | PLAID Dataset | |
|---|---|---|
| | Accuracy (%) | Time(min) |
| AlexNet | 92.86 | 27 |
| VGG16 | 93.50 | 23 |
| GoogLeNet | 94.29 | 20 |
| ResNet34 | 95.48 | 17 |
| CBAM-ResNet34 | 96.38 | 18 |
| Transferred CBAM-ResNet34 | **97.49** | **13** |

## 5.3 Performance comparison of different models

Different models differ in their accuracy of power fingerprint identification. To further illustrate the advantages of the proposed method, AlexNet, VGG16, and GoogLeNet were selected for comparison. The experimental objects all used the color V-I trajectory image of $128 \times 128$ pixels. As shown in Table 2, the transferred CBAM-ResNet34 improved the power fingerprint identification accuracy significantly, and the training time was shorter than that of the above model. The comparison of the identification accuracy of the above-mentioned three models verifies the validity of CBAM-ResNet34 and the model transfer in power fingerprint identification.

## 5.4 Performance comparison of different data balancing algorithms

From Table 1, in the PLAID dataset, Class 5 has 248 samples, while Class 10 has only 73 samples. To solve this problem, it is usually necessary to expand some samples so that the number in different categories is the same. To solve the data imbalance problem in V-I trajectory identification, Class-Balanced(CB) loss is introduced, which is achieved by inverse weighting the loss function by the number of effective classes. Using transferred CBAM-ResNet34 as the base model, the methods in this paper were compared with the original data, random oversampling, SMOTE [41], and SVMSMOTE [42] data balancing algorithms on the PLAID dataset, respectively. Use the F1-score as an evaluation criterion. The experiment was repeated five times to take the mean value and the experimental results are shown in Table 3.

**Table 3. The F1 for different data balance algorithms using transferred CBAM-ResNet34.**

| | Original data | SMOTE | Oversampling | SVMSMOTE | CBAM-ResNet34 |
|---|---|---|---|---|---|
| AC | 0.896 | 0.901 | 0.871 | 0.939 | 0.940 |
| Bulb | 0.976 | 0.956 | 1.000 | 1.000 | 1.000 |
| CFL | 0.933 | 0.926 | 0.921 | 0.946 | 0.993 |
| Fan | 0.946 | 0.955 | 0.963 | 0.913 | 0.989 |
| Fridge | 0.895 | 0.912 | 0.984 | 0.986 | 0.914 |
| Hairdryer | 0.926 | 0.936 | 0.961 | 0.967 | 0.980 |
| Heater | 1.000 | 0.988 | 0.908 | 0.985 | 1.000 |
| Laptop | 0.932 | 0.947 | 0.935 | 0.965 | 0.988 |
| Microwave | 0.951 | 1.000 | 0.983 | 0.985 | 0.978 |
| Vacuum | 0.939 | 0.942 | 0.962 | 0.964 | 1.000 |
| Washing | 0.928 | 0.930 | 0.926 | 0.929 | 0.965 |
| Ave F1 | 0.938 | 0.944 | 0.946 | 0.961 | **0.977** |

**Table 4. The comparison of the proposed method and other power fingerprint identification methods.**

| Load Signature | classification algorithm | Accuracy(%) | $F_{score}$(%) | Time(min) |
|---|---|---|---|---|
| EFD Simplified V-I trajectory | RF | 77.30 | 77.60 | 2 |
| Color V-I trajectory | AlexNet | 92.86 | 93.02 | 27 |
| Power + color V-I trajectory | BPNN | 94.06 | 94.40 | 23 |
| GM-V-I trajectory | Improved AlexNet | 95.08 | 95.36 | 20 |
| Improved color V-I trajectory | Transferred CBAM-RestNet34 | **97.46** | **97.69** | **13** |

From Table 3, we can see that the other three methods can effectively improve the F1-score compared with the original data for training. The method in this paper can significantly improve the F1-score of a few sample categories, such as washing machines and vacuum cleaners, compared with the other three methods, the class-balanced loss is used to weight the Softmax maximum cross-entropy loss (SM-CE) function to increase the weight of minority class samples in the target loss, which makes the classifier focus more on the minority class features in training.

## 5.5 Performance comparison of different methods

To compare the proposed method with existing methods, the critical differences in the load signature, training model, identification accuracy, and other aspects are listed in Table 4. Experiments were conducted on PLAID datasets. Experiment 1 used V-I trajectory features simplified by elliptic Fourier descriptors(EFD), and the classification algorithm was a random forest. Experiment 2 uses color V-I trajectory as loading features, and the classification algorithm is AlexNet. In Experiment 3, the color V-I trajectory features and power features of the loads were fused to form composite features by feature fusion, and the classification algorithm adopted was BPNN. Experiment 4 color encoded the V-I trajectories using the gramian matrix with the improved AlexNet classification algorithm.

As can be seen from Table 4, Experiment 1 only relied on harmonic characteristics for classification and failed to use other features for auxiliary identification. Its identification accuracy was the worst despite the short training time. The color V-I trajectory image adopted in Experiment 2 has a low resolution, and many details are lost. Its identification effect on heaters and other multi-state loads is poor. The load characteristics used in Experiment 3 include the power characteristics of the load, which solves the drawback of losing the power characteristics of the V-I trajectory, but the recognition accuracy and training time is to be further improved. The physical information used in Experiment 4 is the loading characteristics at stable cycles and does not include the loading characteristics at non-stable periods. In the case of color coding using the gramian matrix, each element of the matrix needs to be operated once, which is a relatively large computational effort. Compared to the above experimental methods, the method in this study was improved by feature extraction and model training. Color-encoded V-I trajectory images and transferred CBAM-ResNet34 were used as the load feature and training model, respectively, which significantly enhanced the identification ability of the algorithm and reduced the model training time. Therefore, the load identification effect of the method in this study was better than those of the other four methods.

## 6 Conclusions

This paper proposes power fingerprint identification based on the improved V-I trajectory with color encoding and transferred CBAM-ResNet34. The performance of the algorithm was

verified with the PLAID dataset. Compared with the traditional method, the new method has the following advantages:

- The color encoding is used to construct color V-I trajectory images with different channels in high-dimensional space with higher differentiation. The current, instantaneous power, and trajectory momentum information comprehensively and three-dimensionally reveal the essential properties of various types of electrical equipment and improve the identification accuracy of power fingerprints.

- The CBAM module is introduced and combined with the model transfer method to construct the ResNet34-CBAM model. The CBAM module can amplify the weights of effective channels in the feature layer and fully extract the relevant features of V-I trajectories. The model transfer reduces the training time required for model training and better improves the accuracy of power fingerprint identification in the case of small load data of specific customers.

- The CB loss is introduced to reweight the SM-CE loss function to solve the problem of data imbalance in V-I trajectory identification. Compared with other data balancing methods, the improved class-balanced CBAM-ResNet34 in this paper has better classification performance in small-sample unbalanced NILM datasets.

The experimental results show that the method can better improve the accuracy of power fingerprint identification on small sample imbalanced datasets and reduce the time of classification compared with other advanced identification methods. This validates the effectiveness of the method proposed in this paper.

However, the method proposed in this paper still has some problems to be solved. The identification effect of multi-state load in this paper can be further improved, and more advanced power fingerprint identification models are needed to solve the problem in the future. In addition, the actual engineering implementation faces difficulties in constructing sample libraries and high data sampling frequency requirements. Therefore, a lot of related research work is still needed.

## Supporting information

**S1 Data.**
(XLSX)

## Author Contributions

**Conceptualization:** Lin Lin, Jie Zhang.

**Data curation:** Jiancheng Shi.

**Investigation:** Cheng Chen.

**Methodology:** Jie Zhang.

**Software:** Jie Zhang, Jiancheng Shi, Cheng Chen.

**Writing – original draft:** Lin Lin, Jie Zhang.

**Writing – review & editing:** Xu Gao, Nantian Huang.

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
