## [Decision Letter · Decision Letter 0]

22 Dec 2022

PONE-D-22-31416Power fingerprint identification based on the improved V-I trajectory with color encoding and transferred CBAM-ResNetPLOS ONE

Dear Dr. Lin,

Thank you for submitting your manuscript to PLOS ONE. After careful consideration, we feel that it has merit but does not fully meet PLOS ONE’s publication criteria as it currently stands. Therefore, we invite you to submit a revised version of the manuscript that addresses the points raised during the review process.

We look forward to receiving your revised manuscript.

Kind regards,

Mohamed Hammad, Ph.D.

Academic Editor

PLOS ONE

Journal Requirements:

3. Our internal editors have looked over your manuscript and determined that it is within the scope of our Smart Energy Systems Call for Papers. The Collection will encompass the latest research in smart grid technologies, including information technologies, device integration, distribution methods, and data mining, all towards improving the efficiency of energy supply networks. Additional information can be found on our announcement page: https://collections.plos.org/call-for-papers/smart-energy-systems/. If you would like your manuscript to be considered for this collection, please let us know in your cover letter and we will ensure that your paper is treated as if you were responding to this call. If you would prefer to remove your manuscript from collection consideration, please specify this in the cover letter.

"This work was supported by Natural Science Foundation of Jilin Province (YDZJ202101ZY89)."

6. Please upload a new copy of Figures 1, 3, 4, 9, 10, 11 and 13 as the detail is not clear. Please follow the link for more information:

https://blogs.plos.org/plos/2019/06/looking-good-tips-for-creating-your-plos-figures-graphics/

https://blogs.plos.org/plos/2019/06/looking-good-tips-for-creating-your-plos-figures-graphics/

Reviewers' comments:

Reviewer's Responses to Questions

**Comments to the Author**

1. Is the manuscript technically sound, and do the data support the conclusions?

Reviewer #1: Yes

Reviewer #2: Yes

2. Has the statistical analysis been performed appropriately and rigorously? 

Reviewer #1: N/A

Reviewer #2: Yes

3. Have the authors made all data underlying the findings in their manuscript fully available?

Reviewer #1: Yes

Reviewer #2: Yes

4. Is the manuscript presented in an intelligible fashion and written in standard English?

Reviewer #1: Yes

Reviewer #2: Yes

5. Review Comments to the Author

Reviewer #1: Summary:

In this work, a fingerprint identification based on voltage-current (V-I) trajectory with color encoding and transferred CBAM-ResNet34 is proposed. To obtain a color V-I trajectory image, the current, instantaneous power, and trajectory momentum information are added first to the original V-I trajectory image using color coding.

Then, the ResNet34 model was pretrained using the ImageNet dataset and a fully-connected layer. The Convolutional Block Attention Module (CBAM) was added to each residual structure module of ResNet34. Finally, to solve the problem of data imbalance in V-I trajectory identification, Class-Balanced (CB) loss is introduced to reweight the Softmax cross-entropy (SM-CE) loss function.

The manuscript is interesting; however, the following comment should be addressed :

Abstract :

- - - - - - - - - - -

1 – Results in terms of improvement ratio between the proposed and existing work need to be included .

Introduction Section :

- - - - - - - - - - - - - - - - - - - - - -

2 – This section is fine. No comments .

Materials and methods Section :

- - - - - - - - - - - - - - - - - - - - - - - - - - - - - -

3 – This section is fine. No comments .

Power fingerprint identification based on transferred CBAM-ResNet34 Section :

- - - - - - - - - - - - - - - - - - - - - - - - - - - - - - - - - - - - - - - - - - - - - - - - - - - - - - - - - - - - - - - - - -

4 – This section is fine. No comments .

Experiments Section :

- - - - - - - - - - - - - - - - - - - - - - - - - -

5 – This section is fine. No comments .

Result and discussion Section :

- - - - - - - - - - - - - - - - - - - - - - - - - - -

6 – This section is fine. No comments .

Conclusion Section :

- - - - - - - - - - - - - - - - - - - - - -

7 – The limitation of this work should be clearly included in the conclusion section .

General Comments:

- - - - - - - - - - - - - - - - -

8 - There are some grammatical errors should be corrected .

- - - - - - - - - - - - - - - - - - - - - - - - - - - - - - - - - - - - - - - - - - - - - - - - - - - - - - - - - - - - - - - - - - - - - - - - - - - - - - - - - - - - - - - - - - - - - - - - - - - - - - - - - - - - - - - - - - - - - - - - - - - - - - - - - - - - - - - - - - - - - - - - - - - - - - - - - - - - - - - - - - - - - - - - - - - - - - - - - - - - - - - - - - - - - - - - - - - - - - - - - - - - - - - - - - - - - - - - - - - - - - - - - - - - - - - - - - - - - - - - - - - - - - - - - - - - - - - - - - - - - - - - - - - - - - - - - - - - - - - - - - - - - - - - - - - - - - - - - - - - - - - - - - - - - - - - - - - - - - - - - - - - - - - - - - - - - - - - - - - - - - - - - - - - - - - -

Reviewer #2: Dear Authors

The paper titled “Power fingerprint identification based on the improved V-I trajectory with color encoding and transferred CBAM-ResNet” proposed a power fingerprint identification

based on the improved voltage-current(V-I) trajectory with color encoding and transferred CBAM-ResNet34. First, the current, instantaneous power, and trajectory momentum information are added to the original V-I trajectory image using color coding to obtain a color V-I trajectory image. Then, the ResNet34 model was pretrained using the ImageNet dataset and a new fully-connected layer meeting the device classification goal was used to replace the fully-connected layer of ResNet34. The experimental results on the imbalanced PLAID dataset verify that the method in this paper has better classification capability in small sample imbalanced datasets.

The paper is interesting however it needs improvements.

1. There are various types of ResNet models are available including ResNet-18, ResNet-34, ResNet-50, ResNet-101, ResNet-110, ResNet-152, ResNet-164, ResNet-1202, why authors choose ResNet-34.

2. Line 371-372: the SMOTE is not explained, I suggest the author to explain it with proper work e.g. [1].

3. Line 36: Power fingerprint identification is a hot issue in the field of NILM, include the full form when any abbreviation appears first time in the entire manuscript.

4. This paper used ResNet however the role of ML/DL is not discussed to improve the need to utilize the ML/DL.

5. Introduction needs a new paragraph which discuss the AI/DL applications with different domains at once. This paragraph should be added as “Nowadays, scientists and researchers used the machine learning (ML) and Deep learning (DL) models in several applications including agriculture [1, 2], environment [3-6], and power fingerprint identification. 1. Planetscope Nanosatellites Image Classification Using Machine Learning; 2. CNN Based Automated Weed Detection System Using UAV Imagery; 3. SMOTEDNN: A Novel Model for Air Pollution Forecasting and AQI Classification; 4. CDLSTM: A Novel Model for Climate Change Forecasting; 5. Deep Learning Based Modeling of Groundwater Storage Change; 6. Deep Learning Based Supervised Image Classification Using UAV Images for Forest Areas Classification.

6. Figure 1-13 needs improvement, all these figures looks blur.

6. PLOS authors have the option to publish the peer review history of their article (what does this mean?). If published, this will include your full peer review and any attached files.

Reviewer #1: No

Reviewer #2: No

---

## [Author Response · Author response to Decision Letter 0]

22 Jan 2023

Response to Reviewer 1 Comments

Point 1: Abstract: Results in terms of improvement ratio between the proposed and existing work need to be included.

Response 1: Thank you for underlining this deficiency. In the revised manuscript (page 1), we have added more details about the contribution of the paper in the abstract as the following: 

‘’The experimental results show that the method effectively improves the identification accuracy by 4.4% and reduces the training time of the model by 14 min compared with the existing methods, which meets the accuracy requirements of fine-grained power fingerprint identification.’’

Point 2: Conclusion Section: The limitation of this work should be clearly included in the conclusion section.

Response 2: Thank you for underlining this deficiency. In the revised manuscript (page 23), we have added more details about the limitation of this work in the conclusion section as the following: 

‘’However, the method proposed in this paper still has some problems to be solved. The identification effect of multi-state load in this paper can be further improved, and more advanced device identification models are needed to solve the problem in the future. In addition, the actual engineering implementation faces difficulties in constructing sample libraries and high data sampling frequency requirements. Therefore, a lot of related research work is still needed.’’

Point 3: General Comments: There are some grammatical errors should be corrected.

Response 3: Thank you for underlining this deficiency. In the revised manuscript (page 4, page 7, and page 21), we have corrected grammatical errors and made detailed changes throughout the text. In addition, changes suggested by another reviewer and the grammar of the paragraphs added to the introduction were checked. The correction of major grammatical errors as the following: 

‘’For V-I trajectory classification, the small amount of load data for specific user results in a poorly trained model with low accuracy or low generalization capability [21].’’

‘’(a)The instantaneous voltage of the Fridge over 1 s. (b) The instantaneous current of the Fridge over 1 s. (c) The instantaneous power of the Fridge over 1 s.’’

‘’Color-encoded V-I trajectory images and transferred CBAM-ResNet34 were used as the load feature and training model, respectively, which significantly enhanced the identification ability of the algorithm and reduced the model training time.’’

Response to Reviewer 2 Comments

Point 1: There are various types of ResNet models are available including ResNet-18, ResNet-34, ResNet-50, ResNet-101, ResNet-110, ResNet-152, ResNet-164, ResNet-1202, why authors choose ResNet-34?

Response 1: We are grateful for the suggestion. According to the reviewer’s comment, we have provided more details to describe the possible reasons. We have made additions in the corresponding sections of the manuscript(page 9, and page 11). The reasons for selecting ResNet-34 to construct the classification model in this paper as the following:

The different numbers of ResNet networks represent the number of layers of the network. Compared to ResNet-18, the network with the least number of layers lacks certain feature representation abilities. Compared to ResNet50, ResNet-101, ResNet-110, ResNet-152, ResNet-164, and ResNet-1202, the increase in the number of layers of the network is accompanied by an increase in the number of parameters and computations, which makes the network training slower.

 Also, with the addition of the CBAM attention module, ResNet34 can achieve not only higher accuracy but also higher accuracy at the beginning of training. This indicates that the network can learn the target features more efficiently and speed up the training of the network. This is since the introduction of the attention mechanism inside the residual block can learn the features of V-I image samples more efficiently.

Compared with other layers of ResNet networks, the accuracy improvement of the attention mechanism module on the network is inferior to that of ResNet34. This is because the attention mechanism is more likely to learn effective features in shallow networks, and the improvement effect of the attention mechanism module on the performance of deeper networks is less obvious, and it is difficult for the attention mechanism to learn effective features.

Therefore, considering the hardware cost limitation and the difficulty of training V-I images, and to reflect the feature extraction capability and convergence speed of ResNet34-CBAM, ResNet34-CBAM is selected as the classification model for V-I images in this paper.

Point 2: Line 371-372: the SMOTE is not explained, I suggest the author explain it with proper work e.g. [1].

Response 2: The authors thank the reviewer for pointing out this problem. The required explanation has been made. In the revised manuscript (Line 388, page 19), the revised contents are as follows: 

‘’Using transferred CBAM-ResNet34 as the base model, the methods in this paper were compared with the original data, random oversampling, SMOTE [41], and SVMSMOTE [42] data balancing algorithms on the PLAID dataset, respectively.’’

[41] Kim K. Noise Avoidance SMOTE in Ensemble Learning for Imbalanced Data. IEEE Access. 2021; 9: 143250–65. https://doi.org/10.1109/ACCESS.2021.3120738

[42] Rajesh L, Satyanarayana P. Evaluation of Machine Learning Algorithms for Detection of Malicious Traffic in SCADA Network. J Electr Eng Technol. 2022 Mar; 17(2): 913–28. https://doi.org/10.1007/s42835-021-00931-1

Point 3: Line 36: Power fingerprint identification is a hot issue in the field of NILM, include the full form when any abbreviation appears first time in the entire manuscript.

Response 3: Thank you for underlining this deficiency. In the revised manuscript (Line 39, page 2), the revised contents are as follows: 

‘’Power fingerprint identification is a hot issue in the field of Non-intrusive load monitoring (NILM), which relies on power fingerprint features and classifiers to identify different types of devices. Common power fingerprint characteristics typically include voltage, current, harmonics, power, V-I trajectory, etc [7-9].’’

Point 4: This paper used ResNet however the role of ML/DL is not discussed to improve the need to utilize the ML/DL.

Response 4: We deeply appreciate the reviewer’s suggestion. According to the reviewer’s comment, we have added more details to describe and discuss the role of ML/DL. In the revised manuscript ( page 3), the revised contents are as follows:

‘’Nowadays, scientists and researchers used machine learning (ML) and deep learning (DL) models in several applications including agriculture [15, 16], environment [17-20], and power fingerprint identification. Machine learning is often applied to feature extraction and classification of power fingerprints, such as k-nearest neighbors, support vector machines, decision trees, and random forests. These methods are less computationally intensive, but the identification correct rate is lower. Recently, deep learning has achieved good results in the field of power fingerprint identification, such as CNN, RNN, etc. Meanwhile, researchers propose to construct V-I trajectory images with the help of color coding methods to convert power fingerprint identification into an image classification task in which deep learning excels. However, compared to machine learning, deep learning-based classifiers rely on large-scale training data and longer training time, which limits the application of deep learning.’’

Point 5: Introduction needs a new paragraph which discuss the AI/DL applications with different domains at once. This paragraph should be added as “Nowadays, scientists and researchers used the machine learning (ML) and Deep learning (DL) models in several applications including agriculture [1, 2], environment [3-6], and power fingerprint identification. 1. Planetscope Nanosatellites Image Classification Using Machine Learning; 2. CNN Based Automated Weed Detection System Using UAV Imagery; 3. SMOTEDNN: A Novel Model for Air Pollution Forecasting and AQI Classification; 4. CDLSTM: A Novel Model for Climate Change Forecasting; 5. Deep Learning Based Modeling of Groundwater Storage Change; 6. Deep Learning Based Supervised Image Classification Using UAV Images for Forest Areas Classification.

Response 5: We are grateful for the suggestion. According to the reviewer’s comment, we have provided more details to discuss the AI/DL applications with different domains at once in the introduction. In the revised manuscript ( page 3), the revised contents are as follows:

‘’Nowadays, scientists and researchers used machine learning (ML) and deep learning (DL) models in several applications including agriculture [15, 16], environment [17-20], and power fingerprint identification. Machine learning is often applied to feature extraction and classification of power fingerprints, such as k-nearest neighbors, support vector machines, decision trees, and random forests. These methods are less computationally intensive, but the identification correct rate is lower. Recently, deep learning has achieved good results in the field of power fingerprint identification, such as CNN, RNN, etc. Meanwhile, researchers propose to construct V-I trajectory images with the help of color coding methods to convert power fingerprint identification into an image classification task in which deep learning excels. However, compared to machine learning, deep learning-based classifiers rely on large-scale training data and longer training time, which limits the application of deep learning.’’

[15] Anul Haq M. Planetscope Nanosatellites Image Classification Using Machine Learning. Computer Systems Science and Engineering. 2022; 42(3): 1031–46. https://doi.org/10.32604/csse.2022.023221

[16] Anul Haq M. CNN Based Automated Weed Detection System Using UAV Imagery. Computer Systems Science and Engineering. 2022; 42(2): 837–49. https://doi.org/10.32604/csse.2022.023016

[17] Attaallah A, Ahmad Khan R. SMOTEDNN: A Novel Model for Air Pollution Forecasting and AQI Classification. Computers, Materials & Continua. 2022; 71(1): 1403–25. https://doi.org/10.32604/cmc.2022.021968

[18] Anul Haq M. CDLSTM: A Novel Model for Climate Change Forecasting. Computers, Materials & Continua. 2022; 71(2): 2363–81. https://doi.org/10.32604/cmc.2022.023059

[19] Anul Haq M, Khadar Jilani A, Prabu P. Deep Learning Based Modeling of Groundwater Storage Change. Computers, Materials & Continua. 2022; 70(3): 4599–617. https://doi.org/10.32604/cmc.2022.020495

[20] Haq MA, Rahaman G, Baral P, Ghosh A. Deep Learning Based Supervised Image Classification Using UAV Images for Forest Areas Classification. J Indian Soc Remote Sens. 2021 Mar; 49(3): 601–6. https://doi.org/10.1007/s12524-020-01231-3

Point 6: Figure 1-13 needs improvement, all these figures looks blur.

Response 6: Thank you for underlining this deficiency. In the revised manuscript, we have adjusted the clarity of Figures 1-13.

---

## [Decision Letter · Decision Letter 1]

25 Jan 2023

Power fingerprint identification based on the improved V-I trajectory with color encoding and transferred CBAM-ResNet

PONE-D-22-31416R1

Dear Dr. Lin,

We’re pleased to inform you that your manuscript has been judged scientifically suitable for publication and will be formally accepted for publication once it meets all outstanding technical requirements.

Kind regards,

Mohamed Hammad, Ph.D.

Academic Editor

PLOS ONE

Additional Editor Comments (optional):

Reviewers' comments:

Reviewer's Responses to Questions

**Comments to the Author**

1. If the authors have adequately addressed your comments raised in a previous round of review and you feel that this manuscript is now acceptable for publication, you may indicate that here to bypass the “Comments to the Author” section, enter your conflict of interest statement in the “Confidential to Editor” section, and submit your "Accept" recommendation.

Reviewer #1: All comments have been addressed

Reviewer #2: All comments have been addressed

2. Is the manuscript technically sound, and do the data support the conclusions?

Reviewer #1: Yes

Reviewer #2: Yes

3. Has the statistical analysis been performed appropriately and rigorously? 

Reviewer #1: Yes

Reviewer #2: Yes

4. Have the authors made all data underlying the findings in their manuscript fully available?

Reviewer #1: Yes

Reviewer #2: Yes

5. Is the manuscript presented in an intelligible fashion and written in standard English?

Reviewer #1: Yes

Reviewer #2: Yes

6. Review Comments to the Author

Reviewer #1: Summary:

In this work, a fingerprint identification based on voltage-current (V-I) trajectory with color encoding and transferred CBAM-ResNet34 is proposed. To obtain a color V-I trajectory image, the current, instantaneous power, and trajectory momentum information are added first to the original V-I trajectory image using color coding.

Then, the ResNet34 model was pretrained using the ImageNet dataset and a fully-connected layer. The Convolutional Block Attention Module (CBAM) was added to each residual structure module of ResNet34. Finally, to solve the problem of data imbalance in V-I trajectory identification, Class-Balanced (CB) loss is introduced to reweight the Softmax cross-entropy (SM-CE) loss function.

The authors have addressed the raised comments .

Abstract :

- - - - - - - - - - -

1 – The abstract is fine. No further comments .

Introduction Section :

- - - - - - - - - - - - - - - - - - - - - -

2 – This section is fine. No further comments .

Materials and methods Section :

- - - - - - - - - - - - - - - - - - - - - - - - - - - - - -

3 – This section is fine. No further comments .

Power fingerprint identification based on transferred CBAM-ResNet34 Section :

- - - - - - - - - - - - - - - - - - - - - - - - - - - - - - - - - - - - - - - - - - - - - - - - - - - - - - - - - - - - - - - - - -

4 – This section is fine. No further comments .

Experiments Section :

- - - - - - - - - - - - - - - - - - - - - - - - - -

5 – This section is fine. No further comments .

Result and discussion Section :

- - - - - - - - - - - - - - - - - - - - - - - - - - -

6 – This section is fine. No further comments .

Conclusion Section :

- - - - - - - - - - - - - - - - - - - - - -

7 – This section is fine. No further comments .

General Comments:

- - - - - - - - - - - - - - - - -

8 – No further comments .

- - - - - - - - - - - - - - - - - - - - - - - - - - - - - - - - - - - - - - - - - - - - - - - - - - - - - - - - - - - - - - - - - - - - - - - - - - - - - - - - - - - - - - - - - - - - - - - - - - - - - - - - - - - - - - - - - - - - - - - - - - - - - - - - - - - - - - - - - - - - - - - - - - - - - - - - - - - - - - - - - - - - - - - - - - - - - - - - - - - - - - - - - - - - - - - - - - - - - - - - - - - - - - - - - - - - - - - - - - - - - - - - - - - - - - - - - - - - - - - - - - - - - - - - - - - - - - - - - - - - - - - - - - - - - - - - - - - - - - - - - - - - - - - - - - - - - - - - - - - - - - - - - - - - - - - - - - - - - - - - - - - - - - - - - - - - - - - - - - - - - - - - - - - - - - - - - - - - - - - - - - - - - - - - - - - - - - - - - - - - - - - - - - - - - - - - - - - - - - - - - - - - - - - - - - - - - - - - - - - - - - - - - - - - - - - - - - - - - - - - - - - - - - - - - - - - - - - - - - - - - - - - - - - - - - - - - - - - - - - - - - - - - - - - - - - - - - - - - - - - - - - - - - - - - - - - - - - - - - - - - - - - - - - - - - - - - - - - - - - - - - - - - - - - - - - - - - - - - - - - - - - - - - - - - - - - - - - - - - - - - - - - - - - - - - - - - - - - - - - - - - - - - - - - - - - - - - - - - - - - - - - - - - - - - - - - - - - - - - - - - - - - - - - - - - - - - - - - - - - - - - - - - - - - - - - - - - - - - - - - - - - - - - - - - - - - - - - - - - - - - - - -

Reviewer #2: Dear Authors

I have now completed the review of the revised manuscript, titled " Power fingerprint identification based on the improved V-I trajectory with color encoding and transferred CBAM-ResNet”. I have observed that the authors put in good efforts to address most of the comments satisfactorily. The improvements in the figures is also seems excellent. Ref 17 details are incorrect, please correct.

Best wishes

7. PLOS authors have the option to publish the peer review history of their article (what does this mean?). If published, this will include your full peer review and any attached files.

Reviewer #1: No

Reviewer #2: **Yes: **Dr Mohd Anul Haq

---

## [Editor Report · Acceptance letter]

30 Jan 2023

PONE-D-22-31416R1 

Power fingerprint identification based on the improved V-I trajectory with color encoding and transferred CBAM-ResNet 

Dear Dr. Lin:

I'm pleased to inform you that your manuscript has been deemed suitable for publication in PLOS ONE. Congratulations! Your manuscript is now with our production department. 

Kind regards, 

on behalf of

Dr. Mohamed Hammad 

Academic Editor

PLOS ONE